A specimen of Rhamphorhynchus with soft tissue preservation, stomach contents and a putative coprolite

Hone David 1 dwe_hone@yahoo.com
Henderson Donald M. 2
Therrien François 2
Habib Michael B. 3
1 School of Biological and Chemical Sciences, Queen Mary University of London , London , United Kingdom
2 Royal Tyrrell Museum of Palaeontology , Drumheller, Alberta , Canada
3 Keck School of Medicine, University of Southern California , Los Angeles, CA , United States of America
De Baets Kenneth
Electronic publication date: 2015 Aug 20
Publication date: 2015
Volume: 3
Electronic Location ID: e1191
Received 2015 Mar 29; Accepted 2015 Jul 26
Copyright: © 2015 Hone et al.
Copyright year: 2015
Copyright holder: Hone et al.
License: This is an open access article distributed under the terms of the Creative Commons Attribution License, which permits unrestricted use, distribution, reproduction and adaptation in any medium and for any purpose provided that it is properly attributed. For attribution, the original author(s), title, publication source (PeerJ) and either DOI or URL of the article must be cited.
License URL: https://creativecommons.org/licenses/by/4.0/

Keywords: Rhamphorhynchoid, Palaeoecology, Pterosauria, Rhamphorhynchinae

Funding: Project Daspletosaurus The work was supported by donations via Experiment.com and contributions of numerous donors to Project Daspletosaurus. The funders had no role in study design, data collection and analysis, decision to publish, or preparation of the manuscript.

==============================
Despite being known for nearly two centuries, new specimens of the derived non-pterodactyloid pterosaur Rhamphorhynchus continue to be discovered and reveal new information about their anatomy and palaeobiology. Here we describe a specimen held in the collections of the Royal Tyrrell Museum of Palaeontology, Alberta, Canada that shows both preservation and impressions of soft tissues, and also preserves material interpreted as stomach contents of vertebrate remains and, uniquely, a putative coprolite. The specimen also preserves additional evidence for fibers in the uropatagium.

Introduction

Rhamphorhynchus is a derived non-pterodactyloid pterosaur known primarily from the Late Jurassic ‘plattenkalk’ beds of the Solnhofen region in southern Germany. It is one of the best known and most well represented of pterosaurs and is known from over 100 specimens, many of which are complete and articulated. This includes specimens preserved in three dimensions, and those that have extensive soft tissue preservation (see Wellnhofer, 1975; Frey et al., 2003; Hone, Habib & Lamanna, 2013). The derived Cretaceous pterodactyloid Pteranodon is a rival for this title, being known from many more specimens (in excess of 1,000), many are only isolated elements or fragmentary remains, but soft tissues are unknown (Bennett, 2001). Certainly Rhamphorhynchus is the best known of the non-pterodactyloid pterosaurs, and as such presents a useful study model for many aspects of pterosaur research and has been central to many studies of various aspect of pterosaur biology (e.g., Bennett, 1995; Bennett, 2007; Bonde & Christiansen, 2003; Claessens, O’Connor & Unwin, 2009; Henderson, 2010; Prondvai et al., 2012).

Pterosaur research is perhaps on the cusp of a revolution with a rapid growth in the number of specimens recovered, research and understanding of the clade (Hone, 2012a). As a result, rarely preserved features such as wing membranes or stomach contents are vital to reconstructing the ecology and behavior of pterosaurs, even if they are present from otherwise well-studied taxa. The diet of pterosaurs in particular is controversial and difficult to reconstruct (e.g., see Humphries et al., 2007; Tütken & Hone, 2010; Ősi, 2011; Witton & Naish, 2015) and trophic interactions are key to our understanding of the ecology and behavior of these animals. Despite a wealth of complete specimens, and the often exceptional nature of the preservation, direct evidence of trophic interactions based on stomach contents remains exceptionally rare for pterosaurs. Rhamphorhynchus has commonly been interpreted as being piscivorous based on the long, anteriorly directed and conical teeth, their presence in aquatic systems (Wellnhofer, 1975), and most convincingly, several specimens seen to have consumed fish (Wellnhofer, 1975; Unwin, 2005; Frey & Tischlinger, 2012; Hone, Habib & Lamanna, 2013).

Despite a long history of research and discovery, new specimens of Rhamphorhynchus continue to be discovered with specimens heralding from ongoing excavations (e.g., Frey & Tischlinger, 2012), specimens in collections that had not previously been described (e.g., Hone, 2012b) or those which have been residing in private collections before becoming available to researchers. Here we describe a new specimen of Rhamphorhynchus (TMP 2008.41.001— Fig. 1) that was recently acquired by the Royal Tyrrell Museum of Palaeontology in Alberta, Canada. This preserves extensive impressions of soft tissues, stomach contents of a vertebrate, and a putative coprolite.

Figure 1 Specimen TMP 2008.41.001 of Rhamphorhynchus muensteri.

Scale bar is 100 mm.

Locality Information

Solnhofen, Schernfeld quarry, from Bavaria, Southern Germany.

Systematic Palaeontology

Pterosauria Kaup, 1834	
Rhamphorhynchidae Seeley, 1870	
Rhamphorhynchus Von Meyer, 1847	
R. muensteri Goldfuss, 1831	

Here we follow Bennett (1995) in considering all Solnhofen specimens of Rhamphorhynchus to belong to a single species, R. muensteri. The genus has previously been split into a dozen or more species but these have convincingly been shown to consist of juveniles and subadults of a single species (see Bennett, 1995 for a review). Bennett (1995) provided a strong diagnosis for R. muensteri with numerous autapomorphies, though several of the characters are also present in the recently named Bellubrunnus (Hone et al., 2012). TMP 2008.41.001 clearly belongs to Rhamphorhynchus as it possesses the following features defined by Bennett (1995) not seen in Bellubrunnus (Hone et al., 2012): ten teeth in the upper jaw and seven in the dentary, anterior teeth long and angled anteriorly and laterally, posterior teeth shorter and more vertical, upper temporal fenestra rounded, femur shorter than humerus. Two remaining characters used by Bennett (1995) to define the genus cannot be observed in the specimen: lower temporal fenestra narrow and smaller than the upper, and fourth premaxillary tooth larger and more lateral than other premaxillary teeth. The former cannot be observed owing to the orientation of the skull, but given the size and shape of the upper temporal fenestra is likely correct, and the latter may be the result of intraspecific variation or taphonomic distortion or a temporary condition during tooth growth or replacement.

Description

Specimen TMP 2008.41.001 was purchased from Pangea Fossils Ltd. and brought to the Royal Tyrrell Museum in February of 2008. Notes in the TMP database for the specimen state that it was originally discovered in the Schernfeld Quarry in 1965 and held in possession of the family that owned quarries around the Eichstätt area. This is one of a number of quarries in the Solnhofen basin to have yielded Rhamphorhynchus specimens, but these were not common in Schernfeld, with only five having previously been recorded by Bennett (1995). Comparison of TMP 2008.41.001 with measurements of material in Wellnhofer (1975) suggests that this is not one of the privately held or lost specimens Wellnhofer had seen and so, aside from a single small illustration of the specimen and a brief mention of the tail structure in Persons & Currie (2012), all observations and measurements of this specimen should be new to the scientific literature.

Rhamphorhynchus is known from over 100 specimens and is thoroughly described and illustrated in the literature (Wellnhofer, 1975; Wellnhofer, 1978). Therefore this description focuses on details not commonly seen or preserved in other specimens of the genus.

The specimen (Fig. 1) is an animal approximately 990 mm in wingspan measured as the total of the lengths of both humeri, radii, wing metacarpals and all wing phalanges (Table 1). The bones are of a rich, dark brown colour, though some form of glue or preservative has been added to parts of the skeleton. The bone surface is generally well preserved, though some cortex is lost and broken (e.g., left wing phalanges 2 and 3), and may be present in any counterplate that may exist for the specimen, or were removed and/or destroyed during preparation. The matrix is a very pale yellow-white colour, with occasional flecks of darker sediment, and there are some black dendrites around the bones and along cracks in the slab.

Overall the specimen is in very good condition, well-preserved and articulated with some elements or parts of them preserved in three dimensions. The specimen is presented primarily in ventral view as shown by the position of the sternum overlapping various elements and the lack of visible neural spines and zygapophyses on the cervical and dorsal vertebrae. Some disarticulation has occurred with the shoulder girdles and wings having moved slightly from their natural positions and the ribs and gastralia having been somewhat scattered over the chest of the specimen and part of the centre of the chest has much calcite crystal build up under the preserved elements.

The specimen also preserves impressions of both brachiopatagia and a tail vane, and some traces of the uropatagium (these membranes may also contain some fossilized soft tissues—see below). The abdomen preserves gut contents of an indeterminate vertebrate and there is also a pair of masses of material posterior to the pelvis interpreted as a coprolite.

Skull and mandible

The skull is presented in left dorsolateral view and is partially preserved in three dimensions, though is also somewhat crushed and the left side appears a little distorted (Fig. 2). Some sutures in the skull can be tentatively identified but these are mostly not clear, either because they are being obliterated as a result of cranial fusion during ontogeny, or owing to crushing of elements. All fenestrae of the skull on the left side except the lower temporal fenestra are clearly visible. A fragment of bone is visible through the naris which likely represents a part of the palate. The orbit also contains a T-shaped piece of bone ventral to the partial and poorly preserved sclerotic ring, and this is mostly likely a separated ectopterygoid of the palate.

Figure 2 Photograph of skull showing the ring-like atlas at the rear of the skull and palatal element sitting inside the ventral part of the orbit.

Scale bar is 50 mm.

The left ramus of the mandible is seen in left lateral view and is articulated with the skull. Twelve teeth are preserved in the upper jaw and ten in the mandible. In both cases one or two are apparently only very small (left upper jaw alveoli 8 and 9, left dentary alveoli 3 and 5) and may represent either incipient replacement teeth emerging, or are from the other side of the jaws and so only the tips are visible. This would explain the rather higher count here than is normal for Rhamphorhynchus (ten in the upper jaw and seven in the mandible—Bennett, 1995).

Axial skeleton

Rhamphorhynchus is known to have numerous axial elements show pneumaticity based on major cavities in bones coupled with pneumatopores that pierce the cortex (Bonde & Christiansen, 2003). However here, as with many specimens of this genus, no external pneumatopores are visible.

A ring of bone 5 mm in diameter, but only 2 mm in length is visible at the rear of the skull which is interpreted as the separated atlas (Fig. 2). The axis is partially hidden behind the back of the skull and is in left ventrolateral view such at the left lateral process of the neural arch is visible. The rest of the cervical series is complete (six vertebrae) and articulated and seen in ventral view, although the transition between the cervical and dorsal series is hard to identify. The cervicals are approximately square in shape and the left cervical ribs can be seen in articulation suggesting that the left side is marginally more exposed than the right as a result of crushing.

The dorsal series is difficult to observe as this is partially covered by sternal elements, ribs, and gastralia (Fig. 3). Approximately ten dorsals are preserved in an articulated series and are seen in left ventrolateral view.

Figure 3 (A) Close-up of chest cavity and inferred gut contents in the abdominal region. Scale bar is 20 mm. (B) Map of the major elements seen in Fig. 3.

Bony elements are in dark grey and abbreviations are as follows: cdv, caudal vertebrae; cv, cervical vertebrae; dr, dorsal rib; dv, dorsal vertebrae; fe, femur; hu, humerus; il, ilium; ish, ischium; mc, metacarpal; r, ribs; sc, scapulocoracoid; sac, sacrum; st, sternum; ti, tibia. Possible stomach contents are in light grey—note that the preservation in this area is poor and parts of the highlighted region consist primarily of calcite. Key areas are the possible vertebrae (v) and the long, thin bony rods (br).

The sacrum is well preserved and consists of four vertebrae (Fig. 4). This is very slightly displaced (clockwise in ventral view) relative to both the distal end of the dorsal series and the proximal end of the caudal series. The sacral ribs are broad and fused to the ilium. The tail is preserved in left ventrolateral view as shown by the presence of the transverse process being obscured on the first two vertebrae and the asymmetric presentation of the elongate chevrons and zygapophyses. The divisions between the vertebrae are difficult to distinguish along the majority of the length of the tail and parts are covered by the left pes, so a vertebral count is not possible. Proximally, several of the elongate vertebrae have disarticulated somewhat and are not closely appressed to the caudal centra. The last six preserved caudals are very small (typically around 0.5 mm in length, though the terminal caudal appears to be just 0.05 mm long) and collectively are the same length as the last unreduced caudal (4 mm in total). These tiny terminal caudals are simple and lack the zygapophyses and chevrons of the other the caudal vertebrae, and nor are they bounded by these extensions of the preceding vertebrae.

Figure 4 Close-up of hindlimbs and associated region.

Abbreviations as per Fig. 3, with the following additions: cp, coprolite; ps, pes. Scale bar is 10 mm.

Numerous dorsal ribs and gastralia are preserved on the specimen but a count is not possible given that many elements overlap one another. Many are disarticulated however, and their exact original associations and positions cannot be fully determined. In particular, a number of gastralia are displaced anteriorly and lie below the base of the cervical series (Fig. 3). One sternal rib is preserved adjacent to the anteriormost dorsals and the left margin of the sternum, and shows the typical form of these elements (see Claessens, O’Connor & Unwin, 2009), which are rarely preserved. Several other sternal ribs are preserved alongside the dorsal vertebral column, and two or three further ones are positioned posterior to the sternum but none of these are well preserved.

Pectoral girdle and forelimbs

The sternum is preserved and close to a natural position, with the left hand margin partly overlapping the proximal dorsal centra. The entire right wing, including the right pectoral girdle, has moved as an articulated unit to a position where it lies close to the pelvis. A part of the left scapulacoracoid is preserved close to its natural position but is mostly hidden by the sternum. The right scapula and coracoid are seen in ?anterior view and appear to be nearly fully fused together into a single unit, the visible anterior edge of which has suffered some damage. The distal part of the coracoid is partially buried in the matrix and is underneath a large calcite crystal.

Both wings are well preserved and are nearly completely articulated (Figs. 5 and 6), although the wing phalanges have rotated along their long axes relative to the proximal parts and the right wing metacarpal has separated from the radius and ulna. Both humeri are preserved in medial view, though the right is partially concealed below the femur. The right radius and ulna are better preserved than the left, but the proximal and distal ends of the right are concealed beneath the right humerus and proximal wing phalanx. The left carpals are present and apparently fused into the proximal and distal syncarpal blocks seen in adult pterosaurs, but are poorly preserved and neither pteroid can be seen. In the right wing, the distal parts of metacarpals 1 and 2 are seen having separated slightly from the wing metacarpal and the other elements of the manus are all preserved. In the left wing, only the penultimate phalanx of digit three and all three unguals are visible.

Figure 5 (A) Close-up of the left wing showing preserved membranes. (B) The wing membranes and tail vane are outlined in pale grey and the coprolites are in dark grey.

Abbreviations as above with the following additions: mp, manual phalanges of digits I–IV; ul, ulna, 1–4, wing phalanges 1–4. Scale bar is 100 mm.

Figure 6 (A) Close-up of the right wing showing preserved membranes. (B) The wing membranes and tail vane are outlined in pale grey and the coprolites are in dark grey.

Abbreviations as above with the following additions: mp, manual phalanges of digits I–IV; ul, ulna, 1–4, wing phalanges 1–4. Scale bar is 100 mm.

Both wing fingers are present and articulated, although each wing finger has rotated about its long axis and lies 180° out of the position relative to the proximal parts of their respective wings. In the right wing, the extensor tendon process is fully fused to wing phalanx 1. Both the left and right fourth wing phalanges are moderately posteriorly curved as seen in many pterosaur specimens, including a number of examples of Rhamphorhynchus (Hone, Habib & Lamanna, 2013), and these also terminate in a squared-off tip. In the right wing, the very tip of the fourth wing phalanx is slightly broken, however, there is a clear impression of the tip and this, like the left, is clearly blunt.

Pelvic girdle and hindlimbs

The pelvis is partially disarticulated and some elements appear to have been lost. Both ilia are articulated with the sacrum and appear to be fused to it. The anterior wings of the ilia are well preserved, though the posterior wings are damaged and poorly preserved. The proximal part of the right pubis is articulated with the right ilium, but only the articular end is visible and the rest appears to be hidden below other elements. Only one ischium (?right) can be identified and this is not articulated with, or fused to, the ilium or pubis, but instead has moved anteriorly and lies close to the sternum. The left pubis cannot be seen and appears to be the only major element lost from the specimen. Both prepubes are preserved but are in poor condition and covered by other elements. They are in close association but are not articulated with one another and lie posterior and ventral to the sacrum.

Both hindlimbs are complete and articulated though the right foot is partially hidden under the right wing and the last phalanges of the left foot are hidden by the tail. The midshaft of the right femur is also partially concealed by the right humerus, but the outline of the bone is visible.

Soft tissues

A number of soft tissues or their impressions are preserved in the specimen but in places it is difficult to separate between the two possibilities. These are either impressed into the matrix or raised above it, suggesting they are genuine features and not carved into the matrix artificially, or are the remnants of preparation marks etc. Both brachiopatagia are present (Figs. 5 and 6) and in a relatively natural position (i.e., have not dissociated and moved as seen in some pterosaur specimens cf. Elgin, Hone & Frey, 2011) and are preserved as very faint transparent outlines on the matrix. Each wing has a more narrow chord along most of its length than seen in some specimens of Rhamphorhynchus (e.g., BSPG 1938 I 503a, the ‘Dark Wing’ specimen—Frey et al., 2003) suggesting some postmortem shrinkage of the membranes (Elgin, Hone & Frey, 2011). Both brachiopatagia also appear to have a near 90° turn in them level with the distal end of the radius and ulna, and then become narrower towards the elbow and body. This is likely because the medial part of the wing (the tenopatagium) has fewer or no actinofibrils compared to the more distal part (the actinopatagium) and thus has less support and a greater tendency to shrink or decay after death. Proximal to the elbow, the right tenopatagium (Fig. 6) is rather less clearly preserved than the left actinopatagium (Fig. 5), but does appear to meet the left ankle as is considered common, or even ubiquitous, for pterosaur wing membranes (Elgin, Hone & Frey, 2011).

Under low angle lighting, both actinopatagia show evidence of actinofibrils, though these are considered most likely to be impressions of the fibers, rather than actual preserved soft tissues, since the preserved wings are all but identical in colour to the underlying matrix and are not carbonized or darker than the matrix as in most Solnhofen pterosaurs that preserve soft tissues of the wing (e.g., BSPG 1938 I 503a, the ‘Dark Wing’ specimen, YPM 1778), and are more similar to other specimens considered to be preserved as impressions (e.g., BSPG 1880 II 8).

Identification of the actinofibrils in the matrix is difficult given the very shallow indentations of their preservation, and this is compounded by the fact that the wing membranes have shrunk from their original form. In addition, there must also be some folding to the membrane given the rotation of the wing fingers and the fact that the distal membrane can be seen on both the leading and trailing sides of the right wingtip. Furthermore, at least some parts of the wings have been covered with some form of transparent preservative and brush marks (e.g., swirls) are clearly visible in places on the matrix. Some actinofibrils are visible on the distal left wing and lie subparallel to one another and the fourth wing phalanx as seen in other pterosaurs, including Rhamphorhynchus (Bennett, 2000; Frey et al., 2003). The number and density of the actinofibrils cannot be determined as they are too few and too poorly preserved (perhaps because they are impressions). Lying at approximately 45° to the fibrils in the right wingtip are a series of short apparent grooves (these are approximately 0.3 mm in diameter and 1 mm in length) which are interpreted as small folds or wrinkles in the membrane because these are considerably larger than the typical diameter of actinofibrils in this genus (0.05 mm—Bennett, 2000).

The tips of the wing membranes appear to meet the distal ends of the fourth wing phalanx at an acute angle, and do not show the anteroposteriorly enlarged tips to the membranes as in other pterosaurs (including BSPG 1880 II 8). This may be as a result of postmortem shrinkage, or in the case of the right wing, a result of the folding of the membrane in conjunction with the rotation of the wing finger. The right wing membrane appears on both sides of the fourth phalanx as a result of the rotation of the wing along its long axis.

No part of the propatagium from either arm can be seen, but this is perhaps not surprising given that the pteroids are hidden or lost, and the way the arms have folded might also conceal these membranes even if preserved. Despite the poor preservation around the posterior part of the sacrum and the overlapping elements of the tail and pes, part of the uropatagium, or an impression of it, is preserved (Fig. 7). In the crux of the left hindlimb there is a series of very fine parallel striations running anteroposteriorly and parallel to the tibia, that in gross form matches those seen in the actinopatagia. However, there are unlikely to be stray actinofibrils from the wings given that the wings overall are intact and the tenopatagium, which, although less well preserved than the actinopatagium, would have few or no actinofibrils (Bennett, 2000). Pycnofibers (body covering fibers, that may also be present on the wings—see Kellner et al., 2009) are also not preserved elsewhere on the specimen, and the fibers seen here in the uropatagium are generally too long, thin and straight to be pycnofibers (cf. Kellner et al., 2009). There are also no stray fibers on other parts of the slab, further suggesting that these are genuine and part of the uropatagium.

Figure 7 Photograph taken under binocular microscope of fibrils in the uropatagium.

The large element on the right is the right tibia. Scale bar is 1 mm.

As with the actinofibrils, the fibers in the uropatagium are considered to be preserved primarily as impressions in the matrix, rather than true soft tissue preservation (although this alternative is not ruled out). The clearest part of the uropatagium is perhaps part of the trailing edge because it lies at the very distal end of the tibia (Fig. 8), suggesting a termination close to the ankle as seen in other pterosaurs (e.g., Sordes PIN 2585/3, Pterodactylus, BSPG 1937. I.18). At the distal edge of the uropatagium, a high number of fibrils can be seen to be parallel to the tibia and are associated with a pale yellow stain on the matrix. The individual fibers are approximately 0.06–0.1 mm in diameter, and although their length is difficult to identify, one at least is around 3 mm in length. These are densely packed, with around 12 fibers per mm of membrane (Fig. 8).

Figure 8 (A) Photograph taken under binocular microscope of fibrils in the uropatagium. (B) Interpretive drawing of the uropatagium (pale grey) and fibrils (black lines).

Not all fibrils are illustrated and the width of the lines may not be representative. Scale bar is 2 mm.

Additional striations are visible on the lateral edges of the two tibiae and left metatarsals. These might be scratch marks from preparation, but this seems unlikely as these are in places soft tissues might be expected (decayed uropatagium, proximal tenopatagium, foot webbing) and the marks are very fine and very closely packed and parallel which seems unlikely to be generated by a preparator. Nor do they appear in areas around the skull or anterior to the leading edges of the wing fingers where preparation might be similar to that around the hindlimbs, and nor do they match marks made during preparation of the midsection to reveal the gut contents (done by the TMP in April, 2013). Finally, some of the striations of the uropatagium track across the uneven surface of the matrix (where the yellow staining lies— Fig. 8) suggesting these are not preparation scratches, but impressions tracking the surface of the matrix, and they are not associated with the preservative on the wings noted above, so are not brush marks. These then are most likely fibers of some form but their origin is not clear. The uropatagium has become displaced relative to the bones even in some exceptionally preserved specimens (e.g., Sordes PIN 2585-33). This may be a continuation of the uropatagium but displaced and visible lateral to the tibia.

A diamond-shaped tail vane (or an impression) is also preserved as a near-transparent stain on the specimen, though the dorsal side is preserved as a slight impression, and the ventral side is slightly raised above the level of the matrix (Fig. 6). The vane in total is 61 mm long and has a maximum height of 39 mm. The distal end of this corresponds almost exactly with the tip of the very last reduced caudal of the tail. Very faint impressions of fibers are seen in the tail vane but these are sparse and difficult to separate from the apparent preparation scratches on the surface of the matrix. The fibers are of similar diameter to the impressions of actinofibrils in the distal parts of the brachiopatagia, and are aligned dorsoventrally (i.e., perpendicular to the caudals) in the vane.

The keratinous sheathes of several unguals are also preserved on the specimen as dark orange stains. These are present on the unguals of the right manus and the ungual of digit 1 of the left foot. The claw of manual digit 3 also includes a ‘claw spike’ that is approximately 1.5 mm in length. This spike is a kind of very thin and needle-like extension of the very tip of an ungual. Such a feature is seen on a number of Mesozoic ornithodirans (e.g., the azhdarchid SMNK PAL 3830, and the dromaeosaurid dinosaur Microraptor, IVPP V 1335—D Hone, pers. obs., 2012 and 2010, respectively ) and extends off the tip of the bony ungual and may form part of the ungual, or be an additional element. The lack of the presence of claw sheathes and the keratinous ramphotheca may be the result of loss during preparation.

Finally there are some orange stains around the body of the specimen, which may represent decayed or modified soft tissues. Similar orange soft tissue stains are seen in other Solnhofen Rhamphorhynchus specimens (e.g., CM 11429, NMINH F 10172) and this inference here is supported by the orange colouration of the preserved claw sheathes (goethite stains). There are however, some other orange stains on the matrix not directly associated with the bony parts of the animal, but whether these may represent decayed and drifted organic tissues of the pterosaur, other organic remains, or some geological artefact is not clear.

Ingested material

Gut contents consisting of indeterminate vertebrate elements are preserved in the thoracic cavity of the specimen (Fig. 3). These elements are bounded by the ribs and other thoracic parts of the skeleton (i.e., the bones of the pterosaur lie both above and below the elements in question) and they do not conform in size or shape to any of the elements of the Rhamphorhynchus (and apparently only the pteroids and one part of the pelvis are missing from the specimen). Given the overall articulated and well-preserved nature of the specimen it is unlikely these elements have somehow drifted into this position from outside of the animal and are therefore considered to have been consumed items. A number of these consumed elements are present in the thoracic region which are likely ingested remains of food (i.e., gut contents), but most of these are distorted and difficult to identify though their overall shape appears to be that of squat cylinders. Their exact identity cannot be determined as they are incomplete and partially covered by other elements, and much of the chest cavity has calcite crystal buildup.

A putative coprolite is also preserved in association with the specimen (Fig. 9). This lies almost immediately posterior to the sacrum and thus in a position likely close to the cloaca in life. This has split in two, but the terminal ends of the separated pieces are largely straight and they are of the same size and shape, suggesting a single mass that split along a weak point, rather than two separate pieces. The smaller part (that is closer to the pterosaur’s pelvis) is poorly preserved and shows calcite crystals and is 11 mm long and 3 mm across. The second mass is 8 mm long and 4 mm across and consists of many tens of small and pale comma-shaped or spike-like elements (Fig. 10). These are typically around 0.2–0.3 mm in length, though larger ones are 0.45 mm. Some tiny ones are around 0.05 mm, and are more simple in shape, but these may be partially concealed under other elements as they only appear in the greatest concentration of these pieces.

Figure 9 Detail of the second part of the coprolite.

The series of elements to the left are from the left pes. Scale bar is 5 mm.

Figure 10 Details of the ‘hooklets’ within the coprolites.

Scale bar is 1 mm.

Table 1 Measurements of the major elements of TMP 2008.41.001.

Element or series	Maximum length (to the nearest mm)	
Skull	91	
Cervical vertebrae	53	
Dorsal vertebrae	52	
Sacrum	13	
Caudal vertebrae	259	
Humerus	33	
Radius	62	
Wing metacarpal	19	
Wing phalanx 1	96	
Wing phalanx 2	98	
Wing phalanx 3	93	
Wing phalanx 4	94	
Femur	26	
Tibia	38	

Discussion

Based on the size of the animal and the fusion of various skeletal elements, the specimen TMP 2008.41.001 is considered close to adult status, though there is a mixture of immature and mature characteristics present. In terms of size, it is within the most common range of sizes of elements seen in specimens of Rhamphorhynchus, and these are typically immature (Bennett, 1995) with few adults being known. A number of fused elements show that this animal is not a young juvenile—the scapula is fused to the coracoid and the wing extensor tendon process is fully fused to wing phalanx 1 with an obliterated suture (Figs. 3, 5 and 6). However, the sutures of the skull are still somewhat visible and have not been obliterated as in adults (Fig. 2) and similarly, although the ilium appears well fused to the sacrum, the apparent separation of the pubis and ischium suggest they were not fully fused to each other or the rest of the pelvis (Fig. 4). The bone texture (where well preserved) is smooth and unlike those of very young pterosaurs (Bennett, 1995). Of Bennett’s (1995) year classes for Rhamphorhynchus, the shape of the cranium of TMP 2008.41.001 is intermediate between year class 3 and 4 (and is probably closer to 4), but the mandible matches class 3 well (Bennett, 1995—his Fig. 5). The shape of the tail vane, being a diamond rather than closer to a triangle (as seen in mature specimens), also suggests immaturity (Bennett, 1995, Fig. 6). Collectively then, the evidence suggests that this specimen was not a young juvenile, nor an adult, but the fusion and even obliteration of some sutures in the skeleton, combined with the wingspan and shape of the cranium suggest that it was close to osteological maturity.

The somewhat unusual disarticulation pattern of the specimen is also worthy of comment. The right wing has moved posteriorly, but the ischium has moved anteriorly, as have some of the gastralia. Also the right scapulocoracoid has moved with the right wing, but the left wing is in a natural position, though this has (as a unit), slightly separated from the left scapulocoracoid (Fig. 1). This implies that there was no consistent current or effects of dissociation during decay. The animal presumably came to rest on the substrate on its back, and as the material decayed or was compressed under sediment, collapsed in part to the right, leading to the displacement of the sternum, sacrum and prepubes and perhaps the right wing, and the position of the left leg. Bloating of the carcass during decay may have occurred (Allison & Briggs, 1991) and would explain the anterior movement of the gastralia and the expulsion of the putative coprolite, although the preservation of the wing membranes suggests that there was generally little decay here.

Osteology

The tiny distal caudal vertebrae indicate that most Rhamphorhynchus tails were incomplete, even when they appear to be complete, since the distalmost unreduced caudal may have a rounded posterior face similar to the terminal caudals of many tetrapods. Wellnhofer (1991) had illustrated these tiny vertebrae before, but these are rarely preserved (presumably in part because they are not bounded by the chevrons and zygapophyses) and this feature was overlooked by Lü & Hone (2012) on pterosaur tail lengths. However, as they here constitute less than 1.6% of the total length of the tail, this is unlikely to have any real effect on the data presented to date by Lü & Hone (2012). It does however suggest that similar ‘additional’ caudals may have been present in other pterosaurs but are not often preserved, or may be lost due to careless preparation.

Soft tissues

Despite large numbers of complete and articulated pterosaurs preserved in Konservät-Lagerstätte-type deposits, soft tissues remain rare for pterosaurs, though increasing amounts of material are being discovered and described (Sullivan et al., 2014). The brachiopatagia are probably still the most commonly preserved parts, although some soft tissues that might expect to be commonly preserved are still rare. For example, claw sheathes were first reported for pterosaurs from the Solnhofen in 2003 (Frey et al., 2003) and beaks are also little known, even though they were presumably present on edentulous pterosaurs as well as being known for toothed forms including Pterodactylus and Rhamphorhynchus (Frey et al., 2003). Thus the information preserved here (as preserved tissue and/or impressions) are of importance and do provide corroboration of existing hypotheses.

The part of wing membranes preserved here (Figs. 5 and 6) are most likely the remains of impressions in the matrix, but some soft tissues may be preserved. This is difficult to determine as the wings are seen primarily in ventral view and actinofibrils may be concentrated in the ventral part of the wing (Padian & Rayner, 1993) and can be preserved as natural casts in some specimens. We suggest that these are primarily casts, with the stains representing traces of soft tissues, but this cannot currently be confirmed. The limited extent of the fibers seen in the brachiopatagium may be a result of poor preservation, or because most of the wing is preserved and the fibers are buried within it. Examination under UV light did not reveal any additional details that can be seen under normal natural and artificial lights (see Supplemental Information 1). This does not rule out the presence of soft tissues and imply that the wing membranes are preserved only as impressions as soft tissues do not always fluoresce under UV illumination. Future work with additional lights and filters may therefore reveal additional details. Destructive and/or chemical analysis of the specimen was not considered appropriate given the quality of the material.

The actinofibrils that are seen in the brachiopatagia do conform to the size and shape previously described for these structures in Rhamphorhynchus (and some other pterosaurs) being approximately 0.05 mm in diameter (Padian & Rayner, 1993; Bennett, 2000; Frey et al., 2003) and these conform most closely to the type A wing fibers as described by Kellner et al. (2009).

Confirmation of fibers being present in the uropatagium (Fig. 7) is more important. These have been reported before for pterosaurs, being also present in the holotype of the anurognathid Jeholopterus (Kellner et al., 2009) where fibers are seen to be both subparallel to the long axis of the body and also perpendicular to the tibia as seen here, and fibers of some kind were also suggested for the uropatagium of Eudimorphodon (Wild, 1994). Unwin & Bakhurina (1994) also noted that the scaphognathine Sordes had a large uropatagium replete with fibers, but the size, shape and orientation of these was not discussed. As described above, a series of sub-parallel fibers are present implying the presence of the uropatagium towards the ankles of the animal and imply a typically broad rhamphorhychoid-type uropatagium (e.g., see Unwin, 2005). These are subparallel to the long axis of the body and suggest that fibers did help support the uropatagium in this taxon. Frey et al. (2003) also noted the presence of fibers with the uropatagium in the ‘Dark Wing’ specimen of Rhamphorhycnhus but these were described as being ‘bushy’ and their position on the lateral face of the tibia/fibula suggest these were in fact pycnofibers associated with the body rather than actinofibril-like fibers in the uropatagium itself.

The claw sheathes seen here do seem to be genuinely preserved soft tissues given their clear colour and texture differences to the surrounding matrix. The sheathes are smaller than many described for pterosaurs (e.g., see Frey et al., 2003) as the apparent extent of the sheath extends little beyond the claw-spike of the ungual. However, this may be a result of incomplete preservation, or damage during preparation, and confirmation of short manual claws for Rhamphorhynchus should be sought from additional specimens.

The diet of Rhamphorhynchus

Stomach contents for pterosaurs are very rare, despite the prevalence of these taxa in areas of exceptional preservation that often include soft tissues (e.g., Sullivan et al., 2014). Rhamphorhynchus is perhaps already the genus with the most data in this regard, with several specimens being shown to have elements of fish (Wellnhofer, 1975; Hone, Habib & Lamanna, 2013), or even an entire fish having been consumed (Wellnhofer, 1975; Unwin, 2005; Frey & Tischlinger, 2012). There is little doubt then that, as commonly suggested in the literature (Wellnhofer, 1978; Wellnhofer, 1991; Unwin, 2003; Padian, 2008; Witton, 2013), Rhamphorhynchus was at least occasionally piscivorous.

This interpretation is further supported by the fact that Rhamphorhynchus itself was the victim of attacks by fish (Frey & Tischlinger, 2012) suggesting they were spending significant amount of time over water, and isotope data supports their collecting food from marine systems (Tütken & Hone, 2010), despite likely limitations when at rest on the surface of the water (Hone & Henderson, 2014). The cranial morphology of Rhamphorhynchus and indeed other rhamphorhynchines does appear well suited to taking food from the water with numerous, anteriorly directed teeth and elongate jaws which extend further with a keratinous beak (Frey et al., 2003) as is seen on some modern piscivorous vertebrates and contemporaneous marine predators including a number of plesiosaurs.

Although fish were clearly part of the diet, and Rhamphorhynchus was apparently specialised for taking aquatic prey, this would not rule out other sources of food. Unidentified remains in the stomach of a specimen of Rhamphorhynchus shows that diet was not exclusively fish (Wellnhofer, 1991, p. 160). Carnivorous animals will take animalian food items from well outside their ‘typical’ range if the food is available and there is no reason to think pterosaurs would be different. The specimen here preserves two different traces that in part suggest this genus may have had a diet beyond fish.

First, there are gut contents in the chest cavity of the specimen that are represented by indeterminate vertebrate elements. These bones may represent fish or tetrapod elements, but are not part of the pterosaur as they match none of the dissociated or missing material (ribs, gastralia, sternal ribs, pteroids, pelvic elements) but instead are a subrectangular series and associated subcircular elements that collectively may be vertebrae (Fig. 3). Possible identifications are the opercula of a sizeable fish, or small vertebrae from sharks, though in the case of the former these would be in the absence of all other elements, and the latter implies a more sizeable animal that a small pterosaur may have been able to tackle. Although we cannot absolutely verify the identity of these elements, it is possible that they are tetrapodan—for example in addition to the possibility they represent tetrapodan centra, they also bear a resemblance to some carpals and tarsals of marine crocodilians from the Solnhofen (e.g., Geosaurus). If so, this is the first case of consumption of tetrapodan food items by a pterosaur. Small tetrapods (both aquatic and terrestrial) are known from the Solnhofen (Barthel, Swinburne & Conway Morris, 1990) and of course these would produce still smaller juvenile animals, which would form potential consumed items.

The calcite crystal mass underlying the stomach contents suggests some hard organic matter was originally present because calcite crystals are commonly associated with cartilage in Solnhofen pterosaurs at least (Bennett, 2007). Thus while the only clearly identified remains are the putative vertebrae, the other elements and the calcite mass suggest a sizeable meal was originally present in the digestive tract of the pterosaur.

The part of the inferred coprolite closest to the pelvis of the animal (Fig. 9) is similarly indistinct and apparently consists of a calcite crystal mass again suggesting the presence of harder organic tissues as with the stomach contents. The second part is composed of tiny elements that are simple spikes and hook-like shapes (Fig. 10). These we originally suggested were hooklets from the arms or tentacles of a cephalopod (Hone et al., 2012) but we now tentatively reject this hypothesis as the morphology of the cephalopod hooks are a less good match than we had originally thought. A number of alternatives have also been assessed including the branchial apparatus of a small fish, and possible invertebrate origins such as spines from a small echinoderm or sponge spicules, but none are confident referrals. Examination of the remains of various vertebrates and invertebrates from the Solnhofen (e.g., the teleost fish Gyronchus, the sphenodontid Homeosaurus, the crocodyliform Geosaurus, an indeterminate echinoid and an indeterminate cephalopod—all, D Hone, pers. obs., 2014) do not reveal any compelling matches, but this may be as a result of the unusual preservational situation.

These elements have passed through the digestive tract of an animal and thus will have been affected by digestive processes. They have then been deposited alongside various chemicals and in a fecal mass which would make for a very different local condition to specimens normally preserved in the Solnhofen. Either of these two issues, or both in combination, may have affected the preservational potential of the ‘hooks’ or their appearance and thus identifying them may prove very difficult.

If the diagnosis is correct, this is the first recorded coprolite for any pterosaur. Coprolites are rare for many vertebrate clades (Hunt et al., 2012), and it is likely Rhamphorhynchus defecated over water causing the breakup of the excreted matter. Preservation here is likely as a result of the material being expelled postmortem and in a low energy system thus preventing the dissipation of the fecal pellet. Data from the extant phylogenetic bracket for pterosaurs (crocodilians and birds) and from the digestion and excretion by Mesozoic non-avian dinosaurs is variable. Birds typically produce a near liquid mass, while crocodilians typically produce solid pellets, these can break down quickly (Fisher, 1981). However, a more solid coprolite is known for some birds, and at least one large Mesozoic theropod (Chin et al., 1998). The preservation here of even tiny elements suggests a relatively low amount of acid in the stomach because these have not been destroyed or damaged by their passage (cf. (Fisher, 1981; Andrews & Fernandez-Jalvo, 1998) on crocodilian digestion and waste).

Although we no longer consider the putative coprolite evidence of direct feeding on cephalopods by pterosaurs, this is a plausible hypothesis and worthy of further consideration. Cuttlefish and especially squid match the general form of fish and prey capture would be similar for both, as demonstrated in many modern birds and large predatory fish that may take fish or squid, in addition to some fossil taxa. A diet including both fish and cephalopods has been shown in Larus (gull) species (Baltz & Morejohn, 1977), in pelagic sharks (Lowe et al., 1996) and marine ichthyosaurs (Pollard, 1968). Although some authors may have considered the idea of cephalopods as part of the pterosaur diet implicit in the term ‘piscivory’ it does not seem to have been explicit, even in cases where cephalopods are mentioned. For example, Kemp (2001) noted that both fish and cephalopods would have been in the upper waters of the Solnhofen and local crocodilians would have fed there on both, but despite suggesting pterosaurs would also be limited to feeding in this zone, he suggested they were piscivorous.

Clarification should therefore be made with regards to terms such as ‘piscivory’ to make it explicit the possible prey range encompassed. Both data and analyses of pterosaur diets are increasing (Humphries et al., 2007; Tütken & Hone, 2010; Ősi, 2011; Witton & Naish, 2015) but understanding will be hindered with ambiguous terminology. Even so, the new information here does tentatively suggest a broader diet for pterosaurs than simply fish, and the rapid increase in study in this area is likely to shed additional light on the foraging and feeding behaviour of pterosaurs.

Supplemental Information

Supplemental Information 1 3 photographs of the specimen under UV light

Click here for additional data file.

We thank Brandon Strilisky, Graeme Housego, and Rhian Russell for assistance with access to material held at the Royal Tyrrell Museum. Dawna MacLeod is thanked for the preparation work to further expose elements of the chest and the tip of the tail. Sue Sabrowski is thanked for the UV photographs used in the supplementary files and we thanks Tom Courtenay and Martin Schilling for assisting her. Jim Gardner is thanked for use of his microscope camera. Helmut Tischlinger is thanked for information on the origins of the specimen and we also thank him, Donald Brinkman, Alistair McGowan, David Martill, Lorna Steel, Kenneth De Baets, Andrew Smith, Leititia Adler, Dirk Fuchs, Jakob Vinther, Matthias Mäuser and Andrew Newman for discussions on the identity of the stomach contents and coprolite. Our thanks to Martin Röper for allowing DWEH to access specimens at the Solnhofen Museum. We thank Kenneth De Baets for this work as an editor and also Dino Frey and an anonymous referee for their comments that helped improve the manuscript.

Institutional Abbreviations

BSP Bayerische Staatssammlung für Paläontologie und Geologie, Munich, Germany

CM Carnegie Museum of Natural History, Pittsburgh, Pennsylvania, USA

IVPP Institute of Vertebrate Paleontology and Paleoanthropology, Beijing, China

NMINH National Museum of Ireland, Natural History, Dublin, Ireland

PIN Palaeontological Institute, Russian Academy of Sciences, Moscow, Russia

SMNK Staatliches Museum für Naturkunde Karlsruhe, Karlsruhe, Germany

TMP Royal Tyrrell Museum of Palaeontology, Drumheller, Canada

YPM Yale Peabody Museum, New Haven, USA.

Additional Information and Declarations

Competing Interests

Author Contributions

Donald M. Henderson and François Therrien are employees of the Royal Tyrrell Museum.

David Hone conceived and designed the experiments, performed the experiments, analyzed the data, wrote the paper, prepared figures and/or tables, reviewed drafts of the paper.

Donald M. Henderson conceived and designed the experiments, performed the experiments, analyzed the data, contributed reagents/materials/analysis tools, wrote the paper, reviewed drafts of the paper.

François Therrien conceived and designed the experiments, analyzed the data, contributed reagents/materials/analysis tools, wrote the paper, reviewed drafts of the paper.

Michael B. Habib conceived and designed the experiments, analyzed the data, wrote the paper, reviewed drafts of the paper.

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
