# Peer review of "A specimen of Rhamphorhynchus with soft tissue preservation, stomach contents and a putative coprolite"

_PeerJ, doi:10.7717/peerj.1191_

## Round 0.1 · original submission · Major Revisions

Thanks for submitting your work to PeerJ. The preservation of this specimen is really peculiar and potentially really interesting to interpret pterosaur ecology. I concur with both reviewers, who pointed out the interest of this specimen, but raised several points, which need to be addressed before publication. The main points are:

Description: the description of the skeleton could be improved (see comments by reviewer 1)

Ontogeny: I agree with you that the specimen show characters of both third and fourth ontogenetic classes of Bennett (1995) and would therefore be close to adult. The description and the discussion could however be considerably improved (see comments by myself and reviewer 2).

Taphonomy: I miss a clear consistent taphonomic scenario to explain the peculiar preservation of this specimen. Some isolated statements are made throughout the text, but a more general discussion about the taphonomy of this specimen, would benefit the manuscript and subsequent interpretations made concerning soft-tissue preservation, possible vertebrate food items and a putative coprolite (see further comments by myself and the reviewers)

Soft-tissue preservation or impressions: Please be more consistent and specific. You contradict yourself throughout the manuscript. If you are unsure that these are actual soft-tissues, why did you not test this with additional methods like UV-light or chemical analyses (SEM, etc.)? This should be easy enough to do and would considerably help with the interpretation of your specimen (compare comments by reviewer 2)

Vertebrate prey items: How can you be sure that they are the remains of food items considering the isolated and similar vertebrate remains? And furthermore, how can you rule that these are not just the result of scavenging on dead remains? It would therefore be better to speak of food items instead of prey items as you have not direct support for active predation or scavenging.

Putative coprolite: you mention coprolite in the title, but later in the text you are not entirely sure if this is an actual coprolite. It would therefore be better to use the term “putative coprolite” throughout the text, in the abstract and the title to highlight this uncertainty. I miss a discussion on how can you be sure that is not an other fossil, which is totally unrelated to the pterosaur and accidently associated with it. The interpretation might be considerably improved by UV-light or chemical analyses (SEM, etc.). Compare further comments by myself and the reviewers.

In addition to the suggestions by the reviewers, please also integrate the following points:

page 2, line 21: please rephrase this part to be more in line with the text: “impressions and possible preservation of soft-tissues as well as stomach contents of vertebrate prey as well as a putative coprolite”

page 4, line 16: I would use “putative coprolite” throughout most of the text and in the abstract as you are unsure about it being a coprolite

page 5, line 19-20: you mention that the genus has been split “into a dozen or more species”. By whom? Please cite some references in this context. Later you mention that “these have convincingly been shown to consist of juveniles
20 and subadults of a single species.” By Bennett (1995) or by others ?

page 7, line 22: you mention “cranial fusion”. In this context, it might already be good to hint that the specimen might be close to adult as you discuss later

page 8, line 7-8: you discuss “In both cases one or two are apparently only very small...” Please discuss each cases, separately (one or two).

page 10, line 1: it should be Claessens et al. 2009 instead of Clasessens et al., 2009

page 13, line 4: “wings are preserved” should be replaced by “preserved wings”

page 14, line 20-21: here you say that they are rather impressions that actual soft-tissue preservation, although later on you say you are unsure. It would probably be best to describe them as “soft-tissue impressions, but we can not rule out also some actual soft-tissue preservation in some cases, which we discuss further in detail”.

page 16, line 15-16: “the azhdarchid SMNK PAL 3830, and the dromaeosaurid dinosaur Microraptor, IVPP V 1335”: Please cite references describing these specimens, if these observations have already been made before.

page 17, line 5-10: How can you be sure they are gut contents considering the peculiar state of preservation of your specimen ? I miss a comprehensive discussion of the taphonomic scenario explaining the preservation of this specimen and associated finds. Individual statements considering the taphonomy are scattered throughout the manuscript.

page 17, line 11: How can you rule out that this is another poorly preserved fossil (sponge, etc.) which is unrelated to the pterosaur specimen ? Chemical

page 17, line 17-21: It would be necessary to investigate this with UV-light as well as figure out which chemical composition these elements have, to be able to make further interpretations

page 18, line 13-15: This is a bit contradictory. Please more clearly state if it is an adult or pre-adult or post-juvenile and on what characters this is based on ? Does size have anything to say ? It would just say, that you specimen possesses characters of class 3 and 4 of Bennett(1995) and by which observations these are supported.

Page 19, line 18-22: “Bloating …very little decay here.” Please be more consistent; you say one thing before and turn it around in the next sentence... It would be good to discuss in more detail the taphonomic scenario in a certain passage in the text…

Page 20, line 16: I guess you want to say “Konservät-Lagerstätte-type deposits”

Page 21, line 3-4: please stop contradicting yourself. You say beforehand that they are mainly impressions and to a lesser extent actual soft-tissues...i would therefore turn it around...”probably merely the remains of impressions in the matrix, but they may be some actual soft-tissues there too.”


Page 21, line 10-11: “Examination under UV light may reveal is tissue is genuinely preserved (as hinted at by the orange staining) or merely impressions.” why was this not done? It is fast and easy enough to do and it is actually necessary to clear this question. I would seriously consider to do this as you cannot really resolve this question without it.

Page 22, line 6-10: reviewer 2 pointed out that these authors did refer to these structures differently. Please verify this and cite them more correctly in this context

Page 22, line 19: “In areas of exceptional preservation” I guess this discrepancy is probably related with their ecology? Could you briefly discuss this in this context, for readers not familiar with this aspect.

Page 23, line 10: “piscivorous” instead of “piscovorus”

Page 24, line 1: “subcircular” instead of “subcircualr”

Page 24, line 13: Considering they are just just isolated and always similar elements, how can you rule out scavenging of already partially decayed or disintegrated specimens? It would therefore be better to speak of food items instead of prey items (if the animals they eat were already dead in the case of scavenging instead of preying).

Page 25, line 6: do mean unusual situation or unusual preservation ? I think the latter would be more correct and specific in this context.

Page 25, line 21: Didn´t Hone and Rauhut (2010) discuss the problem with interpretation and rarity of theropod coprolites? It could be cited in this context.

Page 26, line 8-10: “cephaolopds” should be replaced with “cephalopods”

These points can also found in the annotated pdf

Reviewer 1 ·

Basic reporting

-The osteological description is quite short. I know that Rhamphorhynchus specimens are over 100, but among these only a few has been described in detail, mainly emphasizing those anatomical details or differences, which cannot be seen on the specimen described e.g. by Wellhofer. In the description of the present MS only 8 lines deal with the skull. I think at least the openings, their proportions, or some details of the different cranial regions or sutural contacts would be necessary to discuss, especially since most skulls of Rhamphorhynchus are so ossified that the individual cranial elements cannot be identified very well.
To this a detailed technical drawing on the skull would be essential.

- An important part of the MS is the description of the impression or possibly the remnants of the soft-tissue. This is quite good, my only comment is that besides figs. 7 and 8 some more detailed pictures from the fibrils or perhaps technical drawings would be essential to clearly see what the authors think on that.

- Although I cannot see the inferred gut contents within the chest cavity on Fig. 3, I am ok with their conclusions that some of this material in the chest cavity represent the remnants of some bony elements (vertebrae etc). Again, if possible, some more detailed picture on these vertebrate remains would be good.

- Citation-references: Almost complete, vice versa, only a few mispellings (indicated in the pdf) and Kemp (2001) is missing from the Ref list.

- Few comments, questions, suggestions have made directly in the pdf file.

Experimental design

No Comments.

Validity of the findings

The possible coprolite produced by this individual of Rhamphorhynchus would be the really „wow-factor” of this specimen. However, the authors cannot say what are those hooke-like things in that part of the fossil, so it is unambiguous if it is a coprolite or not.
To decide this, it is essential to make some geochemical analysis to clarify the composition of those small hook-like things. I would expect some phosphate content in it, even if those hook-like things are from molluscs or other invertebrates. But at least, the results will help in clarifying what are those things.
An other option would be to use microCT or SEM to see the more details of the inner and/or outer structure.
Based on the figures, it seems that this is only a film-like cover on the slab and not a some kind of 3D stuff that is preserved in the sediment matrix, but I’ve never seen this specimen with a microscope.
Since there is no chemical analysis of the material and you don't know the exact content of this part of the fossil (what are those hook-like things), I would not conclude this so explicitly: „This is the first recorded coprolite for any pterosaur”. (p. 25).

Annotated reviews are not available for download in order to protect the identity of reviewers who chose to remain anonymous.

·

Basic reporting

The article refers to a hitherto unknown species of the long-tailed pterosaur Rhamphorhynchus. The genus is well known.The authors descibe an discuss a new specimen with soft tissue preservation. The critical point is that all evidence is relativated in the text. I have the feeling after reading that the authors either have nothing in hand or they do not really tell it. Before publishing this paper the authors must be clear about the base of their description. Also anatomical direction terms should be used for the description of bone and tissue in order not to get confused with the preservational positions of the bones. Further comments are markes up in the text.

Experimental design

UV examination is missing and the crucial photographs of the uropatagium are blurry and underexposed. Better optics are required here.

Validity of the findings

Big point: basically everything reported in the paper is partly or completely relativated by the authors (s. a.). In clear text this means that each hypothesis presented in the paper is questioned in the next sentrence. Finally the authors seem to have nothing in the hands but suggestions. The fibrills in the uropatagium are by no means proof bases on the quality of the immage. The same hold true for the coprolite, which in the end is only identified by its position. A close-up of the fine structure would be necessary to find out more. This is all distrurbing and declines the valitity of findings drastically. Hence ther would be a potential.

Additional comments

1) refer to the facts, 2) provide positive evidence, 3) enhance the line drawings, 4) provide close-ups for the non-fish vertebrae, 5) provide evidence for the organic presence of a "transparent" membrane, 6) provide real evidence that the soft tissue is not a fake or at least partially not, 7) provide mor infromation on the content of the supposed coprolites.

---

## Round 0.2 · Minor Revisions

Thank you for integrating most of our suggestions. The manuscript has become even more interesting and more easy to follow, but some minor points still need to be addressed:

Personal observations on other specimens or taxa: you make comparisons with various other tetrapods which are based on new, personal observation (as you mention in the rebuttal letter). The manuscript would benefit from mentioning who of you made these observations as well as when and in which context these observations were made as it currently not entirely clear what is based on literature observations and which ones are based on personal observations.

Identification of hook-like elements in coprolite: You know tentatively reject your own hypothesis (Hone et al., 2012) that these elements are hooklets from arms or tentacles of a cephalopod. I agree that cephalopod hooks might not be the correct interpretation, but you need to present the arguments against such an interpretations or which changed your opinion (e.g., shape is very different from typical cephalopods hooks, etc.). Furthermore, you mention comparisons with various other invertebrates and vertebrates, but you need to be more specific with which higher taxa they were compared, particularly those which have become known from the Solnhofen Plattenkalk. I also missed a brief mention of other forms with small needle to hook-like elements (e.g., sponges) in addition to squids, vertebrates or echinoderms.

Comparison with other vertebrates eating both fish and squids: in the rebuttal letter, you mention pelagic birds (including pinguins) and plesiosaurs which regularly eat cephalopods in addition to fish; I feel the manuscript would benefit considerably from briefly integrating this in the discussion of the manuscript with some references.

UV-Light investigation: thank you for applying UV-light to investigate your specimens; it would be great to add some more information on how you did this or a picture illustrating these observations. It would also be great a sentence on why you did not use chemical methods to avoid destructive analysis (as you discuss in the mention). This is a valid reason and might be obvious to you, but not to the reader.

Some typos were still found, which need to be changed before acceptance as those are difficult to alter once the manuscript is accepted. All references were listed.

These and additional comments can be found in annotated pdf

---

## Round 0.3 · Minor Revisions

Thanks for integrating these final recommendations and suggestions. Your manuscript is as good as accepted. Two minor spelling errors are still present within the manuscript (see annotated pdf), which need to be taken care off before I can sent it into production.

---

## Round 0.4 · accepted · Accept

Thank you for making the last corrections.